# A Vision–Language Model-Based Traffic Sign Detection Method for High-Resolution Drone Images: A Case Study in Guyuan, China

**DOI:** 10.3390/s24175800

**Published:** 2024-09-06

**Authors:** Jianqun Yao, Jinming Li, Yuxuan Li, Mingzhu Zhang, Chen Zuo, Shi Dong, Zhe Dai

**Affiliations:** 1CCCC Infrastructure Maintenance Group Co., Ltd., Beijing 100011, China; yaojianqun@126.com (J.Y.); lijm6216@163.com (J.L.); lyx624074264@163.com (Y.L.); 2School of Transportation Engineering, Chang’an University, Xi’an 710064, China; mingzhuzhangzmz@163.com (M.Z.); dongshi@chd.edu.cn (S.D.); zhedai@chd.edu.cn (Z.D.)

**Keywords:** vision–language model, traffic sign detection, drone images, multi-modal learning

## Abstract

As a fundamental element of the transportation system, traffic signs are widely used to guide traffic behaviors. In recent years, drones have emerged as an important tool for monitoring the conditions of traffic signs. However, the existing image processing technique is heavily reliant on image annotations. It is time consuming to build a high-quality dataset with diverse training images and human annotations. In this paper, we introduce the utilization of Vision–language Models (VLMs) in the traffic sign detection task. Without the need for discrete image labels, the rapid deployment is fulfilled by the multi-modal learning and large-scale pretrained networks. First, we compile a keyword dictionary to explain traffic signs. The Chinese national standard is used to suggest the shape and color information. Our program conducts Bootstrapping Language-image Pretraining v2 (BLIPv2) to translate representative images into text descriptions. Second, a Contrastive Language-image Pretraining (CLIP) framework is applied to characterize not only drone images but also text descriptions. Our method utilizes the pretrained encoder network to create visual features and word embeddings. Third, the category of each traffic sign is predicted according to the similarity between drone images and keywords. Cosine distance and softmax function are performed to calculate the class probability distribution. To evaluate the performance, we apply the proposed method in a practical application. The drone images captured from Guyuan, China, are employed to record the conditions of traffic signs. Further experiments include two widely used public datasets. The calculation results indicate that our vision–language model-based method has an acceptable prediction accuracy and low training cost.

## 1. Introduction

As part of fundamental road infrastructures, traffic signs play a key role in ensuring transportation safety and smooth traffic flow [1,2]. Traffic signs provide essential guidance to restrict the behaviors of vehicles, drivers, cyclists and pedestrians. Numerous vehicle crashes are prevented based on crucial information, including the speed limit and warning of potential hazards. Therefore, the maintaining the conditions of traffic signs becomes an important task in infrastructure maintenance [3]. Regular inspections, cleaning and replacements have a positive effect on the visibility and legibility of traffic signs. In recent years, Unmanned Aerial Vehicles (UAVs) and drones have gained considerable attention in the context of infrastructure maintenance [4,5,6]. The main advantages of UAVs include their high-resolution images, flying flexibility and high speeds. UAVs can conveniently span extensive areas. Moreover, the combination of UAV images and computer vision techniques has become a prevalent and active topic. In particular, a variety of the traffic sign detection methods are used to analyze UAV images. Inspired by high-performance platforms, the traffic sign detection program provides a low-cost way to identify the locations and categories of traffic signs. Compared with manual operations, inspection time and use of resources are significantly saved.

Traditional traffic sign detection techniques mainly rely on digital-image processing techniques [3]. The following are the three main approaches: color thresholding, shape detection, and feature extraction. Color thresholding techniques transform images into color spaces. Prior knowledge, including hue, saturation, and brightness, is applied to find traffic signs [7]. Shape detection methods employ edge detection, curve fitting and Hough transform to identify the region of interest. The traffic sign area is distinguished from the background. Feature extraction methods concentrate on the local visual characteristics of the image. The program utilizes image descriptors like the Scale-Invariant Feature Transform (SIFT), Histogram of Oriented Gradients (HoG) and Local Binary Pattern (LBP) to generate an image feature vector [8,9]. Next, these feature vectors are fed into a machine learning-based classifier. The relationship between the input feature vectors and the output category is explored by the supervised learning program. The conventional machine learning program includes decision tree, support vector machines and random forest [10,11].

Since 2013, with the rise of deep learning techniques, neural networks and convolutional neural networks (CNNs) are broadly used in a variety of traffic sign detection scenarios [12,13]. In particular, object detection methods have become a prevailing choice for traffic sign detection tasks [14]. As a core component in computer vision, the object detection program focuses on not only pinpointing the location but also classifying the object under research. Thus, there are two main tasks within the object detection framework, namely localization and classification. 

In general, the deep-learning-based object detection method can be divided into one-stage and two-stage approaches. As a classical one-stage detection method, You Only Look Once (YOLO) uses a convolutional neural network as the backbone module [15,16,17]. In order to accelerate the running speed, YOLO directly maps the input image into the output bounding box. Accordingly, the object detection is fulfilled with a single pass of the deep neural network. On the other hand, the two-stage object detection method pays attention to improving the computational accuracy. For example, there are two primary steps in Region-based Convolutional Neural Networks (RCNNs) [18,19,20]. The first step is to generate numerous candidate bounding boxes. Then, the computer concentrates on classifying these candidate regions. Compared with the one-stage method, the two-stage approach alleviates the misdetection issues and achieves high detection accuracy. Aiming at achieving real-time detection, a collection of lightweight networks can reportedly speed up traffic sign detection. Attention mechanisms and feature aggregation modules are advisable methods for improving efficiency [21,22,23]. In recent years, Detection Transformer (DETR) networks have emerged as a cutting-edge method in the field of object detection [24]. The main contribution of DETR networks is to combine the local visual feature extraction capabilities of convolutional neural networks and the sequence data analysis strength of the Transformer model. In other words, DETR networks not only consider the association between each bounding box and the target instance but also account for the positional relationships between individual bounding boxes. Therefore, the detection quality issue is effectively mitigated.

With the advancements in computation platforms and image acquisition, deep-learning-based traffic sign detection methods have gained a wide range of real-world applications. However, the extensive utilizations of the deep learning techniques encounter several substantial challenges. First, deep neural networks such as YOLO, RCNNs, and DETR networks highly rely on labeled image datasets to learn the non-linear projection between input images and output predictions. In real-world scenarios, it is costly and expensive to manually annotate high-value targets in numerous images. The lack of high-quality datasets not only slows down the convergence of neural networks but also results in the overfitting problem. Second, training and inference are two main steps within the deep neural networks. The performance of the object detection program is considerably impacted by the difference between training materials and real-world testing images. In practice, traffic signs encompass a variety of categories and rich semantic information. It is challenging to build a training dataset covering all potential cases and train a neural network to handle unseen categories. Third, both YOLO and RCNNs use convolutional neural networks as a fundamental component for extracting visual features. Hyperparameter settings, such as learning rate, gradient descent and the number of convolutional layers, have a considerable influence on the performance of neural networks. The parameter specification step often consumes a lot of time and computational resources in the real-world scenario.

Since 2020, vision language models (VLMs) have gained significant attention in the community of artificial intelligence and computer vision [25]. The benefits of VLMs are listed as follows: (1) As a core concept in VLMs and large-scale pretrained models, the self-supervised learning technique is dedicated to directly extracting visual features from images without manually labeled data [26,27]. In training with massive image datasets, the self-supervised techniques significantly reduce the resource consumption associated with image collection and manual labeling. The neural network pretrained by the self-supervised learning technique becomes a favorable starting point for tackling the downstream image analysis tasks. (2) VLMs have a strong multi-modal capability and zero-shot prediction ability. The traditional image classification and object detection programs are highly dependent on discrete labels to express target instances. In the VLM framework, the similarity between visual content and textual explanation can be quantified by comparing feature vectors. The multi-modal ability allows users to customize text descriptions to explicitly explain the semantic meaning of target instances. (3) Vision transformers (ViTs) are broadly used as the basic architecture in most VLMs [28,29,30]. Compared to CNNs, a ViT offers a better image understanding ability and analysis accuracy. The main reason lies in its capability to exploit long-range spatial dependency and global features in input images. Thus, a ViT excels in complex scenarios such as image classification, semantic segmentation and image generation.

In this paper, we explore a vision–language model-based traffic sign detection method. Our research focuses on addressing the training cost issue and slow deployment problem. Within the existing traffic sign detection framework, considerable computation resources are consumed by creating human annotations and fine-tuning a neural network. In addition, the inconsistency between training sets and testing sets can cause difficulties in program deployment. Compared with previous programs, two key points of the proposed method are as follows: (1) As important prior knowledge, the text description is used to depict the traffic signs and guide the image processing program. (2) To solve the training burden, we apply the large-scale pretrained network to analyze the image and text input. The basic steps of our method are listed as follows. First, a keyword dictionary is created to express the semantic meaning of various traffic signs. We referred to the Chinese national standard to obtain the shape and color information of common traffic signs. In addition, the Bootstrapping Language-image Pretraining version 2 (BLIPv2) model is applied to realize the image translation. High-quality text descriptions are automatically produced in accordance with the representative traffic sign images. Second, Contrastive Language-image Pretraining (CLIP) is launched for multi-modal feature extraction. An image encoder is responsible for extracting visual features from UAV images. In comparison, a text encoder is devised to project each element of the keyword dictionary into a high-dimensional language feature vector. Third, we calculate the similarity according to the cosine distance between the visual and textual features. The Softmax function is applied to predict the probability distribution and determine the category of each traffic sign instance. 

We test the proposed method in a highway scenario in Guyuan, China. Accuracy, precision, recall, F1 score and frame per second (FPS) are used to quantitatively evaluate the performance. Furthermore, the performance of the VLM is checked by two public traffic sign datasets. The computation results indicate that our method achieves high classification quality with low resource consumption. Considering its ability, the VLM offers new insights for traffic sign detection research.

The main contributions of our proposed method are summarized as follows:(1).The multi-modal learning technique is introduced to alleviate the reliance on the annotations and solve the domain shift issue. Instead of discrete labels in the annotated dataset, the text description becomes an important material in regard to representing the prior knowledge of traffic signs.(2).In order to save training time, we launch the vision–language model to analyze UAV images as well as text descriptions. There is no fine-tuning step because our program benefits from the strong generalization ability of the large-scale pretrained model. Therefore, the proposed method can be efficiently deployed in a new scenario.(3).The proposed method is applied in a real-world case study in Guyuan, China. Two public datasets are used to compare our program with other milestone methods. The computational results indicate that the proposed vision–language model-based method exhibits competitive performance in terms of generalization ability and cross-domain adaptation. Exploring the multi-modal learning and large-scale network methods may prove to be beneficial in regard to solving the practical traffic sign detection task.

The rest of the paper is arranged as follows: Section 2 provides a brief overview of the previous deep learning methods in the field of traffic sign detection. The technical details of the proposed program are explained in Section 3. Section 4 and Section 5 introduce the practical applications of our method. A group of evaluation metrics are presented to assess the detection performance. Finally, the conclusion is summarized in Section 6.

## 2. Existing Deep-Learning-Based Traffic Sign Detection Method

As a mainstream method in the field of the traffic sign detection, the YOLO neural network is renowned for its excellent inference speed and real-time performance [15]. Figure 1 provides the basic steps of the YOLO-based traffic sign detection program. At first, the researchers and participants depict the bounding boxes on each traffic sign instance. A training image dataset associated with human annotations is created. Then, a training program is launched to train the YOLO network. The YOLO network concentrates on learning the relationship between the input image and output boxes. Finally, the performance of the YOLO network is examined in a practical application. The traffic sign instances are rapidly found in the real-world case. To save computational time and resources, YOLO employs a convolutional neural network to simultaneously realize the localization and recognition tasks. On one hand, the object localization is equivalent to a regression problem. The core goal is to predict not only the coordinates of the object center but also the height and width of the bounding box. On the other hand, the program treats the recognition task as a classification program. The neural network focuses on estimating the possibility of each category.

To further improve the capability of YOLO, researchers continuously report new technical improvements. In 2017, the YOLOv2 network is suggested. Data augmentation, the smoothing L1 loss function and Softmax function are introduced to improve the detection accuracy [31]. To address the concern of the bounding box size, YOLOv2 uses a clustering algorithm to produce anchor boxes from the training dataset. Subsequently, the YOLOv3 network adopts the intersection of union (IoU) loss function and the Focal loss function to enhance object localization and classification quality [32]. To accurately analyze images, YOLOv3 applies Darknet-53 as its backbone module. The increased number of convolutional layers has a positive influence on capturing complex visual information. In 2020, YOLOv4 and YOLOv5 focus on extracting hierarchical image features [33,34]. The path aggregation network, spatial pyramid pooling and feature pyramid network are reported to integrate feature information across different resolutions. These modifications enable YOLO to simultaneously detect large and small objects. In the past three years, YOLOv6, YOLOv7, and YOLOv8 have been proposed [35,36]. Cross-scale training, attention mechanisms and model distillation are incorporated into the YOLO framework. These developments significantly advance the capabilities of one-stage target detection networks and greatly improve the ability of neural networks to detect target instances.

Although the YOLO network has made significant progress in the field of intelligent transportation, the traffic sign detection methods encounter the following several key technical limitations: (1) The mainstream traffic sign detection methods are highly dependent on labeled image datasets. The number of images, the diversity in the dataset and the annotation quality are decisive factors in the recognition accuracy. In practical applications, it is challenging to collect an adequate amount of labeled data for specific scenarios. The lack of training data slows down the neural network convergence and contributes to the overfitting problem. (2) There are two main procedures in object detection: training and inference. Therefore, one key challenge is to tackle the difference between the training dataset and the actual measured dataset. Traffic sign instances have a broad range of styles, colors and shapes. It is difficult to introduce abundant semantic information and appropriate diversity into the training image dataset. (3) Networks such as YOLO and RCNNs use convolutional neural networks as their core backbone module. Hyperparameter settings, including learning rate, optimizer, neural network architecture and image resolution, have a substantial effect on the recognition accuracy. Researchers have to spend considerable time to find the optimal parameter configuration for specific application scenarios. The extensive time requirement brings a difficulty to the rapid and flexible deployment of traffic sign detection programs.

## 3. The Vison Language Model-Based Traffic Sign Detection Method

To realize the rapid deployment of the traffic sign detection, this paper introduces a vision–language model-based approach to deal with high-resolution UAV images. As Figure 2 shows, the proposed method consists of three basic modules. At first, a keyword dictionary is constructed utilizing the Chinese national standard and BLIPv2. Next, we carry out the multi-modal feature extraction. CLIP is activated to generate the visual and textual feature vectors. Finally, the program performs the similarity measure step to determine the category of traffic signs. 

There are two benefits of our method, as follows: (1) The dependence on the annotated dataset is significantly mitigated by the multi-modal learning program. The text description becomes a valuable alternative to the discrete image labels. (2) The fine-tuning step is saved by the employment of vision–language models. The pretrained VLM programs create a solid foundation to analyze the UAV images as well as text descriptions.

### 3.1. Keyword Dictionary Construction to Explain Traffic Signs

The first step of the proposed method in this work is to build a keyword dictionary describing the semantic information of various traffic signs. To generate high-quality text descriptions, we design two approaches in the dictionary construction step. As the first method, it is important to note that the visual characteristics of all traffic signs are strictly constrained with the national regulation. According to Chinese national standards [37], there are six main categories of traffic signs. Each class has its specified combination of the base color, character color, border color, shape and location. Representative images for each category are shown in Figure 3. 

The specific definitions of each type of traffic sign are discussed as follows. (1) Prohibition signs are used to impose strict restrictions on traffic behaviors. Common examples include stop, no vehicles, no pedestrians and no U-turn. Prohibition signs are usually circular in China. In order to provide favorable visibility, the main colors include red, white and black. (2) Mandatory signs are employed to guide drivers as well as pedestrians. Typical examples include turning signs, roundabouts, one-way streets and pedestrian crossings. Mandatory signs are characterized by their circular or rectangular shape, blue background and white symbols. (3) Warning signs promote traffic participants to observe road and traffic conditions. Sharp curves, continuous curves, rockfall areas and crosswinds are common examples of warning signs. In general, triangular shapes with large yellow areas and black characters are used to indicate the presence of dangerous situations. (4) Wayfinding signs provide information on road direction, location and distance. They usually have large areas to convey adequate information. Rectangular borders and white text are main features. The blue and green backgrounds are separately used to imply urban roads and highways. (5) Tourist area signs are responsible for supplying directions and distances to tourist attractions. To improve visibility, they have brown backgrounds and white fonts. (6) Information signs focus on providing information about off-road facilities, safe driving and other relevant information. Typical examples include motorway numbers and seat belt reminders. The main shape is rectangular with white areas and black characters.

Based on the text description in Chinese national standards, we construct an ensemble of English sentences to explain various traffic signs. Figure 4 illustrates English descriptions corresponding to each traffic sign category. In general, five or six sentences are summarized from the explanation in Chinese national standards. It is clear that the color and shape information plays an important role. For example, the phrase ‘a red circular border’ is commonly used to describe the prohibition sign. In general, red is a signal to imply a dangerous situation and offer high visibility. Circles are beneficial in terms of improving clarity and simplicity. In comparison, the words ‘blue’, ‘yellow’, ‘green’, ‘dark red’ and ‘white and black’ are broadly applied to describe the rest of traffic signs. Furthermore, a noteworthy phenomenon is that only three sentences are created to describe the tourist sign. The main reason for this lies in the limited content of the Chinese national standard. Compared with the other five categories, the regulation of tourist signs is not adequate. 

Moreover, we introduce a second way to create the keyword dictionary. Compared with the text description in the Chinese national standard, the example UAV images become explicitly prior material to express the visual characteristics of traffic signs. As Figure 5 shows, we activate the Bootstrapping Language-image Pretraining Version 2 (BLIPv2) program to convert an image into an English description [38,39]. As an excellent vision–language pretraining method, BLIPv2 is devised to connect the objects in images and their language descriptions. Aiming at improving the performance, BLIP applies a unified multi-task learning framework to simultaneously fulfill the understanding-based task as well as the generation-based task. 

There are three primary objectives within the BLIP training program. First, the image–text contrastive learning module focuses on aligning the image representation and text representation in the feature space. As a contrastive learning step, the program is devoted to maximizing the image–text similarities between semantically matching samples across modalities. In comparison, the distance between the image and text is supposed to be large when unrelated instances are provided. Second, the image-grounded text generation step is responsible for generating the text description according to the intrinsic characteristics of an input image. The visual information is converted into the image caption. Using the cross-entropy as the loss function, the text generator attempts to enlarge the likelihood of each word in an autoregressive manner. The label smoothing strategy is utilized to improve the robustness and generalization of the neural network. Third, the image–text matching module is suggested to learn the fine-grained alignment between image and text representation. As a binary classification problem, the image–text matching program is suggested to predict whether an image–text pair is matched or unmatched. The similar instances constitute a positive pair. In comparison, a negative image–text pair indicates that there are two unmatched and incompatible samples. With the intention of developing the prediction accuracy, a bi-directional self-attention mask is employed to bridge images and texts. A two-class linear classifier is used to output the classification result. In addition, several technical modifications are made by BLIPv2 to develop the training efficiency and reduce trainable parameters. A lightweight Querying transformer is devised to bridge the modality gap and facilitate cross-modal interaction. Moreover, BLIPv2 highlights the role of large-scale pretrained neural networks. On one hand, a frozen image encoder network is applied to yield the high-level visual feature in accordance with an input image. On the other hand, the generative language capability is improved by a frozen large language model.

Based on the pretraining step in a massive dataset of images and their captions, BLIPv2 is capable of understanding the visual information as well as generating high-quality text descriptions. In this work, we employ a BLIPv2 program to analyze the traffic signs captured by high-resolution UAV images. In order to avoid the influence of outliers, we assign two representative images into each category. Since there are six classes of Chinese traffic signs, 12 representative images are involved in this case. The English sentences created by BLIPv2 are shown in Figure 5. It is apparent that the color information plays an important role in text description. The words red, yellow, blue and green explicitly present the main characteristics of Chinese traffic signs. 

### 3.2. Multi-Modal Feature Extraction with Contrastive Language-Image Pretraining

After the keyword dictionary construction step, the next technical problem is to analyze the drone image and text description. It is necessary to convert the unstructured image and language data into numerical vectors. The feature vector is also referred to as embedding in the deep learning community. In the feature space, each vector represents an instance. The semantically matching samples share similar positions. In contrast, the large distance between the two points reveals that there is a substantial difference between the two examples. 

Image and text feature extraction programs are active topics in the field of computer vision and natural language processing. The traditional image processing technique introduces a variety of handcrafted descriptors to analyze the input image. For example, Scale-invariant Feature Transform (SIFT) and Histogram of Gradient (HoG) pay attention to calculating the distribution of local gradients. The difference between the template center and neighboring points becomes a decisive element with the local binary pattern (LBP) framework. With the development of deep learning, the neural network becomes a valuable alternative to achieve image feature extraction. After training on a massive dataset, convolutional neural networks (CNNs) and vision transformers (ViTs) have a powerful capability for capturing task-specific and robust embeddings. A high-dimensional feature vector is produced to explain the visual content. On the other hand, the text characterization technique also has experienced the shift from handcrafted descriptor to deep-learning-based method. The traditional text feature extraction techniques include bag-of-words (BoW), term frequency–inverse document frequency (TF-IDF) and N-grams. The frequencies of specific words have a huge influence on feature vectors. Inspired by the success in sequential data processing, recurrent neural networks (RNNs), long short-term memory (LSTM) and transformer networks are broadly used to extract high-level features from the sentences.

In this work, we apply the Contrastive Language-image Pretraining (CLIP) framework to realize the image and text feature extraction [40]. As a prominent method in the field of multi-modal learning, the core concept of CLIP is to create a joint representation space for both images and text. In other words, both visual information and textural descriptions are projected into a unified feature space. The advantages of CLIP are embodied by the following three aspects. (1) A transformer network is used to capture the high-level features from the raw data. There are two major components within CLIP. On one hand, an image encoder pays attention to fulfilling the image feature extraction. A ViT model is applied to project a 2D input image into a 1D feature vector. Rather than the convolution calculation in a CNN, a ViT applies the attention mechanism to explain the interaction between different image patches. Therefore, a ViT is a viable tool for exploring the global structure and extract visual content. On the other hand, the text encoder is responsible for mapping textual descriptions into numerical representations. A transformer-based architecture is utilized to capture long-range dependency within language sentences. The self-attention mechanism in the transformer plays an important role in understanding the correlation between each word. (2) Self-supervised learning is an essential part of the CLIP framework. Aiming at alleviating the reliance on data labeling, there is no human annotation in the training dataset. To promote the network training program, CLIP is dedicated to automatically generating labels of training examples. Motivated by the success of contrastive learning, the relationship between images and text becomes a key concept. On the basis of a massive dataset of image–text pairs, CLIP trains neural networks by encouraging the similar instances to be close together. The image and its corresponding caption share the close positions in the feature space. In comparison, incompatible image and text examples are pushed apart and assigned a large distance. (3) There are 400 million image–text pairs in the training dataset. The diverse examples in this extensive dataset establish a favorable foundation to handle the overfitting problem. Instead of memorizing specific instances, the neural network tends to capture the underlying patterns within the training data. Moreover, a wide range of training data is useful for ensuring generalization and robustness. The image–text pairs with varying conditions help the program to mitigate the data bias and generate an effective representation of different modality.

Based on the preceding discussion, one important procedure is generating visual feature vectors in accordance with the CLIP image encoder network. An illustration is shown in Figure 6. A pretrained ViT program is launched to analyze the drone image. There are four key steps, as follows: First, the input image is evenly partitioned into several non-overlapping patches. In other words, a sequence of image patches is provided to the following steps. In general, a patch of size 16 × 16 or 32 × 32 is used. Second, the program converts each 2D patch into a low-dimension vector on the basis of a reshaping operation and a linear projection layer. This vector is also referred to as patch embedding in many studies. Moreover, the position embeddings are combined with the patch embedding in order to record the location of each patch. Third, a transformer encoder module is designed to deal with a sequence of patch embeddings. While encoder-decoder architecture is widely used in the transformer for natural language processing, a ViT only applies the encoder module to generate the image embedding according to the visual content. Two basic modules are involved in the transformer encoder step. On one hand, the self-attention mechanism concentrates on exploring the correlation between each element in the embedding sequence. On the other hand, a feed forward network conducts non-linear calculations to learn complex structures. Fourth, the multi-head attention module becomes a sensible choice in a ViT in regard to learning a variety of spatial dependencies between image patches. Based on a sequence of patch embeddings, several self-attention modules are carried out in parallel. The outputs of the attention modules are concatenated and fed into the final layer to generate a rich visual representation.

Moreover, our program launches a transformer-based text encoder network to realize the text feature extraction. Similar to a ViT, the attention mechanism plays an important role in the transformer network [41]. The second row of Figure 6 provides the basic steps of the text encoder network. The technical details of the text encoder network are discussed in the following. At first, tokenization is a necessary preprocessing step. The computer divides the input sentence into individual words. The starting and ending markers are predefined to express the structure of a sentence. Next, a pretrained word embedding program is carried out to map the English words into a numerical vector. This vector conveys the basic semantic content of each word. Afterward, the program uses the positional embedding method to preserve the location of each word. Then, a transformer encoder network is performed to quantify the correlation between each two words. Like a ViT, multi-head attention and feed forward networks are essential components in the transformer architecture. Finally, a linear projection layer is launched to generate the resulting output. The embeddings independently created by the attention modules are combined to produce an effective text feature vector.

### 3.3. Traffic Sign Classification with Similarity Measurement

By understanding both the visual information of the drone image and the semantic information of the text description, the computer is able to determine the category of traffic signs. The predicted probability distribution is generated by calculating the similarity between the visual feature vector and the text feature vector. In this paper, our program uses cosine distance as the criterion for similarity calculation. Different from Euclidean distance and Hamming distance, cosine distance focuses on the angle between the input vectors. The influence of vector magnitude is significantly alleviated on the similarity calculation. Therefore, the cosine distance becomes a suitable approach to quantify the similarities between high-dimensional vectors. In addition, the value of cosine distance ranges from 0 to 1. The value of 0 indicates that two input vectors share the same direction. Assuming that we have n-dimensional visual feature vector X and textual feature vector Y, the cosine distance between the two embeddings is defined as follows:(1)$$\mathrm{dis}\left(X,Y\right)=1-\frac{\sum_{i=1}^{n}x_{i}y_{i}}{\sqrt{\sum_{i=1}^{n}x_{i}^{2}}\sqrt{\sum_{i=1}^{n}y_{i}^{2}}}$$
where $x_{i}$ is the i-th element of the feature vector X.

Based on the calculation results of the cosine similarity, our method uses the Softmax function to generate the predicted probability distribution. Based on the discussion in Section 3.1, we produce 33 keywords according to the Chinese national standard. Suppose that the computer focuses on addressing one input image. Thus, one visual feature vector and 33 textual feature vectors are created in this case. In accordance with Equation (1), a distance set $\left(dis_{1},dis_{2},\ldots ,dis_{33}\right)$ is yielded. Then, the predicted probability corresponding to the k-th text description is calculated as follows:(2)$$p_{k}=\frac{e^{dis_{k}}}{\sum_{j}^{33}e^{dis_{j}}}$$

On the basis of the probability distribution, our program selects the text description associated with the maximum probability as the predicted category.

## 4. The Traffic Sign Detection Task: A Case Study in Guyuan, China

### 4.1. Data Collection and Data Preprocessing

In this section, a practical application in Guyuan, China, is used to examine the proposed method. The location of Guyuan is shown in Figure 7. As a city in Northwestern China and Ningxia Province, Guyuan spans over 10,540 square kilometers and has a residential population of 1,142,000 people. Moreover, Guyuan is characterized by a complicated topography, drought climate and low rainfall. Affected by the Jing River and Qingshui River, the landscape of Guyuan consists of large interlaced loess, beams, mountains and trenches. The highway network of Guyuan extends for about 408.6 km. In particular, a highway of 333.6 km in length has been constructed over the past three years. The major nation-level highways include G70 Fuyin, G85Yinkun and G22 Qinglan. Due to the challenging natural environment and a high proportion of bridges and tunnels, infrastructure maintenance in Guyuan has become a difficult task. Therefore, it is crucial to develop an effective detection method that continuously monitors the condition of traffic signs. With the intention of ensuring transportation safety as well as operation efficiency, the damaged signs should be promptly identified and repaired.

In this application, we employ a UAV to investigate the infrastructure condition of highways in Guyuan. An amount of 111 high-resolution images of size 8000 × 6000 pixels are captured to indicate the operating state of the traffic signs. Figure 3 provides representative examples of the traffic signs. It is apparent that there are a variety of imaging circumstances in our UAV dataset. The imaging angles, shooting distance and illumination condition considerably enlarge the diversity of traffic sign datasets. These variations present significant challenges for the subsequent classification programs.

### 4.2. Traffic Sign Detection with VLM and Supervised YOLO Networks

For the 111 UAV images, this study utilizes the CLIP image encoder to extract visual features. Given an input image, the image encoder creates a high-dimensional feature vector. The ViT-B/32 model is chosen as the backbone module. As an important component within the CLIP framework, there are three main parts of ViT-B/32, namely the input embedding layer, the Transformer encoder and the multi-layer perceptron. In ViT-B/32, the total number of learnable parameters is approximately 86 million. The image processing program is explained in detail. (1) The input embedding layer is responsible for partitioning the input image into a series of patches of size 32 × 32. Based on linear mapping operation, the image patches are encoded into 768-dimensional vectors. (2) The transformer encoder concentrates on finding long-range dependencies between high-dimensional vectors and exploring the spatial structures of the image. Furthermore, a multi-head attention mechanism is applied to identify the spatial relationship from various perspectives. Aiming at capturing the fine-grained visual content, 12 encoder modules are adopted by ViT-B/32. (3) As the final component, the multi-layer perceptron is devoted to controlling the dimensionality of the output vector. The program creates a 512-dimensional feature vector to represent the input UAV image. The detailed technical specifications of the ViT-B/32 model are available in the literature [29]. Since 111 UAV images are involved, the CLIP image encoder associated with ViT-B/32 model yields a feature matrix of size 111 × 512 to express the visual content.

Next, the CLIP text encoder is launched to analyze text descriptions created by the Chinese national standard as well as the BLIPv2 model. In this application, the Text Transformer is selected as the backbone network. There are 12 layers within the Text Transformer framework. Each layer has eight multi-head attention mechanisms. Nearly 63 million parameters are involved in this deep neural network. The architecture of the Text Transformer network is detailed in the literature [40,41]. It is worth noting that there are two ways to generate text descriptions in our method. On one hand, 33 sentences are created according to the Chinese national standard. Therefore, a text feature matrix of size 33 × 512 is obtained by the CLIP text encoder. On the other hand, our program views 12 representative images as the prior material to express the six categories of Chinese traffic signs. BLIPv2 is activated to translate images and produce the text description. Therefore, our program yields a text feature matrix of size 12 × 512. 

Based on the feature matrix mentioned above, we calculate the cosine distance as a measure of the cross-modal similarity. The class probability estimation is performed using the Softmax function. In the following paragraph, the drone images in Figure 3 and the text description in Figure 4 are used as the example. The cosine distance between the visual feature matrix and the text feature matrix is shown in Figure 8. The three smallest distances are highlighted in orange. Figure 9 displays the predicted probability distributions. The maximum probability and its corresponding category are emphasized in red.

In order to provide an in-depth understanding, the computation result in Figure 8a and Figure 9a is analyzed. The three keywords with the highest probabilities are “U-turn not allowed”, “a photo of a prohibited traffic sign with a red circular border on the side of a highway” and “a photo of a traffic sign prohibiting vehicles with a red circular border”. It is clear that the first keyword describes the prohibition of U-turn behavior. In comparison, the second and third keywords emphasize the red boundary of the prohibition sign. Figure 9a presents the predicted probability of the traffic sign category for this image. The possibilities associated with three candidate keywords are 20%, 14% and 9%, respectively. Accordingly, the computer identifies the target category in Figure 7a as a prohibition sign. The calculation results in Figure 8 and Figure 9 indicate that the vision–language model method used in this study effectively predict the categories of traffic sign images.

To further evaluate the performance of the designed method, we manually classified all 111 traffic sign images. Based on the human identification, the drone images consist of 23 prohibition signs, 5 mandatory signs, 26 warning signs, 35 wayfinding signs, 13 tourist area signs and 9 information signs. Next, we applied the proposed method to classify the traffic signs in the UAV images. The accuracy and reliability of the recognition method are quantitatively assessed using classification accuracy. In the classification task, the ratio between the number of correct prediction and the total number of samples becomes the evaluation metric. Based on the text description created by the Chinese national standard, Figure 10a displays the number of samples as well as the classification recognition accuracy for each category. The overall classification accuracy is 86%. A noticeable phenomenon is that there are no human annotations in our detection program. During the keyword construction step, the Chinese national standard plays a key role in generating the text description. The drone images are not involved. Therefore, the annotation cost is significantly mitigated by the keyword dictionary.

Next, we analyze the classification results performed by comparing the drone images and text descriptions created by BLIPv2. Given two representative images for each category, BLIPv2 creates 12 English sentences to summarize the main characteristics of Chinese traffic sign. Then, the CLIP text encoder network is carried out to generate the text embeddings. Finally, the classification task is realized by the cosine distance and the Softmax function. The text description with the maximum probability becomes the classification result. The classification result is shown in Figure 10b. Given 111 drone images, the classification accuracy is 89%. This finding indicates that the multi-modal ability of the vision–language model establishes a robust foundation for the traffic sign detection problem.

The computation time for each computation module is discussed in Figure 11. We compile our traffic sign detection program in Python 3. As a prevailing programming library, PyTorch 2.3 is used to implement the deep learning method. Our computer configuration includes Windows 10, a Inter I5-12490F of 3.0 GHz, and a memory of 32 GB. A NVIDIA RTX 4060 is employed to carry out the deep learning computation. It should be noting that we have two keyword dictionaries. In accordance with Figure 4, the first dictionary is created using the Chinese national standard. In comparison, 12 text descriptions are generated by the BLIPv2 program. 

The detailed time consumption is explained as follows. First, the BLIPv2 program attempts to translate 12 representative images into text descriptions. Based on our computing platform, 6.13 s are spent to handle one image. Thus, the dictionary construction step takes 73.56 s to generate appropriate keywords. Second, 1.86 s are consumed to load the multi-modal processing program within the CLIP framework. Third, the computer takes 0.014 s to extract high-dimensional embedding from the 33 sentences generated from the Chinese national standard. In comparison, 0.006 s is necessary to analyze 12 BLIPv2 text descriptions. Fourth, the CLIP image encoder network is launched to extract visual features from a given image. Since there are 111 images in our dataset, 7.35 s are required to create a high-dimensional feature matrix. Therefore, CLIP takes 0.06 s to analyze high-resolution drone images. Fifth, the proposed method utilizes cosine distance and the Softmax function to specify the category of each traffic sign. A time of 0.06 s is taken to deal with 111 drone images. Based on the discussion mentioned above, the overall running time of our vision–language model-based traffic sign detection is 9.29 s.

According to Figure 11 and the preceding time analysis, it is obvious that the high computation efficiency is realized by introducing the vision–language model in the traffic sign detection task. Accordingly, the advantages of the multi-modal traffic sign detection method proposed in this paper include the following: (1) The multi-modal recognition approach significantly reduces the running cost for image labeling and image dataset building. The keyword dictionary is rapidly created based on the Chinese national standard and representative images. In comparison, a high-quality labeled dataset is an important part of the mainstream deep learning frameworks. Substantial labor is consumed to create accurate human annotations. Moreover, noise in the image annotations has a negative effect on the network training procedure. (2) Our vision–language model-based method directly employs a pretrained neural network to fulfill the image and text feature extraction. The need for training and fine-tuning deep neural networks is mitigated to realize the rapid deployment. The pretrained network significantly alleviates the dependence of the deep learning method on high-performance computing resources.

### 4.3. Comparison with Supervised YOLO Networks

To quantitatively assess the performance of the designed method, we implemented YOLOv5 and YOLOv8 networks as the comparison methods. YOLOv5 is an object detection network developed by Ultralytics in 2020 [34]. Compared to the previous methods, YOLOv5 introduces the Focus loss function, path aggregation network and feature pyramid with the intention of developing detection accuracy and efficiency. On the other hand, YOLOv8 is released by Ultralytics in January 2023 [35]. The improved computational accuracy and efficiency is achieved across general-purpose object detection datasets.

As a classical supervised object detection network, the annotated bounding boxes are necessary materials within the YOLO network. In this case, all traffic signs in 111 drone images are identified by the expert. Then, our program divides 111 images into training and testing sets using an 80% to 20% ratio. In other words, the first 88 images are independently adopted to train two YOLO neural networks. The remaining 23 images are used to assess the performance of the deep neural network. To ensure reliable prediction results, we set the number of training epochs as 150 and the batch size for the neural network as 4. The network optimizer and learning rate schedule separately specify the adaptive moment estimation (Adam) and the cosine annealing schedule. The initial learning rate is controlled as 0.001. Based on the dataset configuration mentioned above, YOLOv5 takes 28.2 min to complete the neural network training process. In contrast, the training program in YOLOv8 took 28.8 min. In accordance with the time performance, it is evident that supervised learning frameworks heavily rely on labeled datasets. The neural network is constantly upgraded based on the visual features of a specific dataset. However, the image labeling and neural network training are time-consuming operations.

The trained YOLO models are employed for the traffic sign detection task. Treating 23 images as the testing dataset, the YOLOv5 network correctly predicts 20 traffic signs. However, two traffic signs are not identified. One traffic sign is misclassified. Therefore, the overall classification accuracy of 23 testing images is 86%. On the other hand, 19 traffic signs are accurately classified by the YOLOv8 program. Three traffic signs are neglected and one traffic sign instance is misclassified. Thus, the classification accuracy of YOLOv8 is only 83%. The classification accuracy of the two YOLO networks for various traffic signs is explained in Figure 12. It is evident that two object detection networks suffer from the limited accuracy of mandatory signs, tourist signs and information signs. Only 50% of the classification accuracy is realized when two mandatory signs are fed into the two YOLO networks. The reason for the limited accuracy of the YOLO networks lies in its dependency on the labeled dataset. It is challenging to train a deep neural network when there are a few training images for a specified category. In this scenario, 89 training images and six categories are introduced. However, the number of training images for direction signs, tourist signs and information signs is 3, 9 and 7, respectively. The shortage of training data has a negative impact on the supervised learning method in regard to learning the effective image features.

Next, we apply five common quantitative evaluation metrics to check the proposed method. (1) As a straightforward method, accuracy is defined as the proportion of correct predictions to the total instances. In general, high accuracy indicates that a program has been used favorably in a detection task. However, the accuracy is not effective when the program encounters an imbalanced classification problem. (2) Precision focuses on the ratio of positive predictions created by the machine learning program which are actually correct. A high precision reveals that the program yields a low false positive rate. (3) Recall is devoted to checking the proportion of actual positive instances that have been identified. This metric is important when the cost of false negative is high. (4) As the harmonic mean of precision and recall, the F1 score attempts to find a balanced measure of the detection performance. (5) Frames Per Second (FPS) pays attention to quantifying the running speed. It represents the number of individual images can be processed by the computer program. A high FPS rate implies that the program has a high computational efficiency and real-time capability. The detailed definitions of the preceding concepts are explained in [42].

The computation results of the performance evaluation are exhibited in Table 1. The finding can be explained from twofold aspects. First, our VLM-based method has a comparable detection quality with the supervised YOLOv5. However, the YOLOv8 did not obtain a satisfactory result in this application. The values of accuracy, precision, recall and F1 score in the last row are lower than those in the first three rows. The main reason for this is that a small dataset with limited training images is not sufficient for training a YOLOv8 network. Second, the proposed method has a competitive performance in terms of running speed. Based on the VLM method, 14.98 high-resolution images can be evaluated per second. This running speed is slightly faster than the two YOLO networks. The primary factor behind the high efficiency is that CLIP focuses on understanding the global structure and extracting a visual feature vector from an input image. In comparison, the YOLO networks employ the image partition strategy. In order to explore the local pattern, the input image is uniformly divided into several patches. The high-resolution UAV images lead to an increase in the computation burden.

### 4.4. Bootstrapping Test on the Small Dataset

In the previous section, we used a dataset with 111 drone images to examine our VLM-based method and supervised YOLO networks. However, a small dataset did not establish a solid foundation with the scientific evaluation. Therefore, we apply the bootstrapping test to check the randomness and variability with the traffic sign detection methods. The computer creates multiple new datasets by resampling the original UAV images with replacements. Image datasets with different distributions are fed into the network training procedure. The diversified datasets are helpful for checking the generalization ability of the detection method under various circumstances.

The bootstrapping test comprises three main steps. (1) A group of new training datasets are created by randomly resampling the original dataset. Each bootstrapped dataset has the same size as our UAV image dataset. A key point is that the image is sampled with replacements. This means an image can be selected multiple times in an individual bootstrapped dataset. (2) Based on the bootstrapped dataset, the training procedure is launched to upgrade the deep neural network. It should be noted that we have four programs in the previous section. First, all images in the bootstrapped dataset are used as the test material for the VLM program with keyword dictionary 1. There is no training step since the text description is generated from the Chinese national standard. Second, we utilize the BLIPv2 program to generate the keyword dictionary. For each bootstrapped dataset, 12 text descriptions are produced to depict the traffic signs in the bootstrapped dataset. The traffic sign detection task is realized by calculating the similarities between the drone images and the keyword dictionary. Third, the program trains the YOLOv5 and YOLOv8 networks. The parameter setting is the same as the previous program in Section 4.3. For each bootstrapped dataset, the first 88 images are fed into the training procedure. After the fine-tuning step, the remaining 23 images are applied as the test set within the supervised learning framework. (3) The model performance is evaluated according to the test images in the bootstrapped dataset. In this case, we apply accuracy, precision, recall and F1 score as the evaluation metrics. (4) In order to understand the robustness and stability, we combine the evaluation results from every detection program. The distribution of performance metrics becomes an important indicator of the detection quality. A small range reveals that the traffic sign prediction results are consistent across the bootstrapped samples. The detection program is not significantly influenced by the dataset variation. In comparison, a large deviation implies that there is high variability and randomness within the predictions. The detection program is sensitive to the variation within the image dataset.

Based on the preceding framework, we carry out the bootstrapping test on our drone image dataset. In particular, the number of bootstrap samples is specified as 10. This means 10 new datasets are generated from the original drone images. The distribution of evaluation metrics is shown in Figure 13. It is clear that the VLM with dictionary 1 has an impressive detection result. The detection quality is not substantially affected by the bootstrapped dataset. The main reason is that the text descriptions created using the Chinese national standard provide excellent material for expressing the intrinsic characteristics of traffic signs. In comparison, the detections of the second dictionary are not consistent across multiple bootstrapped datasets. In order to express the target instances, our program launches BLIPv2 to generate text descriptions. Given an input image, the image translation program creates a sentence to describe the main object. Thus, the quality of training images has a substantial influence on the subsequent detection results. The reliance on the specific characteristics of training images enables dictionary 2 to be sensitive to the image variation. Furthermore, a wide distribution is observed when the YOLO networks are performed in this traffic sign detection task. As the supervised learning framework, the quality and diversity are decisive factors in regard to the network convergence. An image dataset with limited training examples makes generating a robust deep neural network difficult. In some cases, the network may overfit to specific patterns or noises in the bootstrapped datasets. The preceding phenomenon reveals that it is a challenging task to train a reliable deep learning model with a small dataset.

## 5. The Application of Traffic Sign Detection on Public Datasets

### 5.1. Experiment Configuration

In this section, two public traffic sign datasets are employed with the aim of extensively evaluating the proposed method. On one hand, we test the performance on Chinese Traffic Sign Detection Benchmark 2021 (CCTSDB2021) [42]. On the other hand, Tsinghua-Tecent 100K (TT100K) is applied to check our VLM program [43]. Since there is no fine-tuning step in our program, we focus on the test sets in CCTSDB2021 and TT100K. Three example images in these two test sets are shown in Figure 14. 

It is clear that there are several significant differences between our UAV dataset and the two public traffic sign datasets. (1) Complex backgrounds are involved in CCTSDB2021 and TT100K. In these two public datasets, the vehicle-based camera is used to capture the traffic sign instance. It is difficult for the detection program to distinguish the visual elements, including vehicles, traffic lights, trees and billboards. In comparison, UAV images are gathered from the aerial platform. The traffic signs are always at the center of UAV images. (2) The image resolution remarkably varies in two public datasets. Notably, 860 × 480, 1280 × 720 and 1920 × 1080 are common image sizes in CCTSDB2021. The testing image is sized 2048 × 2048 in TT100K. High-resolution images of size 8000 × 6000 are collected by the UAV. Thus, the high-resolution image provides sufficient information to realize the fine-grained analysis. (3) The images have experienced considerable visual variations in CCTSDB2021. According to the weather conditions and illumination, the testing images are partitioned into the following six groups: sunny, cloud, night, snow, foggy and rain. In contrast, there are no extreme weather conditions in the TT100K and UAV datasets. (4) The multi-object detection task is a core concept in CCTSDB2021 and TT100K. In general, there are multiple traffic sign examples in the test image. As a comparison, our UAV images concentrate on the single-object detection task. In other words, our program pays attention to identifying the main traffic sign instance within the image. (5) Three categories of traffic signs exist in CCTSDB2021 and TT100K. Prohibition, warning and mandatory sign are annotated according to the semantic meaning. On the basis of the Chinese national standard, six categories are employed in our UAV dataset.

It is worth noting that the primary reason for the difference between our UAV dataset and the two public traffic sign datasets lies in the different research problems. In our scenario, the main task is to monitor the condition of traffic signs in the context of infrastructure maintenance. The UAV technique is explored to capture images from the aerial platform. Accordingly, the traffic sign is always located at the center of UAV images. There are no extreme weather conditions or illumination issues. Despite the relatively simple visual content in UAV images, the core challenge is to fulfill the rapid deployment and realize the cross-domain adaptation. In contrast, the main task in CCTSDB2021 and TT100K is to promote the development of traffic sign detection in the field of autonomous driving. Thus, the vehicle-based camera is used to capture the traffic scenario. A competitive traffic sign detection method is supposed to deal with varying sizes, changing illumination, complex background and multiple objects.

In order to deal with CCTSDB2021 and TT100K, two modifications are conducted on our VLM-based method. (1) A pretrained YOLO network becomes a preprocessing step to localize the traffic signs in two public datasets. The program outputs a group of image patches encompassing the traffic sign instance. Based on the keyword dictionary created by the Chinese national standard, we carry out the CLIP and similarity calculations to predict the categories of investigated examples. (2) There are three specified categories of traffic signs. Our program only employs the keywords describing the prohibition, mandatory and warning signs.

### 5.2. Quantitative Performance Evaluation

In this section, we focus on the performance of the proposed VLM-based method on the two public traffic sign datasets. Similar to Section 4.3, a variety of evaluation metrics are used to assess the performance. Table 2 displays the detection results on two public datasets. Two key observations are found. First, the values of accuracy, precision and recall are larger than 80%. This indicates that the VLM-based program outputs reliable programs. Second, our program uses accelerating behavior to deal with two public datasets. Compared with high-resolution UAV images, the images of moderate size in CCTSDB2021 and TT100K have a positive effect on speeding up the feature extraction step.

Next, the 1500 test images in CCTSDB2021 are further analyzed. Based on Zhang et al. [42], there are two viewpoints in regard to understanding the detection result. On one hand, the traffic sign instances are divided into five subsets according to their size. Access small (XS), small (S), medium (M), large (L) and extra large (XL) are used. On the other hand, six groups are generated on the basis of weather conditions. The extreme weather scenarios include rain, fog and snow. In addition, we introduce the detection results performed by other important detection programs. The computation results are directly cited by Zhang et al. [42,44]. Table 3 and Table 4 exhibit the detection results for CCTSDB2021. In Figure 15, a column chart is created to facilitate the data visualization.

It should be noted that no fine-tuning step is performed in our program. The VLM-based method does not employ the training dataset in CCTSDB2021 and TT100K. Based on the text description created by the Chinese national standard, the computer attempts to determine the categories of traffic signs according to the distance between visual content and language features. The image variations among diverse datasets can create a substantial challenge for the detection program. In contrast, RCNN, YOLOv5, FCOS and one-level feature programs in Table 3 and Table 4 belong to the supervised learning. There are two necessary steps. At first, the computer carries out the fine-tuning step to train the deep neural network. Then, the converged network is launched to identify the traffic signs in the test set. Thus, the effectiveness of the fine-tuning step has a major influence on the detection result.

### 5.3. Discussion on the Detection Result

Based on Table 3 and Table 4, there are two important findings. (1) Our VLM-based method exhibits competitive performance when the test image is taken in favorable environments. Given an input image under sunny, cloud and night conditions, our program has comparable results to the supervised detection programs. The main reason is that the generalization ability of the multi-modal learning method is helpful for dealing with the domain shift and realizing cross-domain adaptation. The visual characteristics of the traffic signs in the UAV dataset, CCTSDB2021 and TT100K are supposed to comply with Chinese regulations. Therefore, the text description provides advisable material to guide the deep learning program. The rapid deployment of traffic sign detection can be realized by exploring the consistency between the image and text description. (2) One shortcoming of the proposed method is that the detection result is highly influenced by the color information. Figure 16 displays several false detections. In our text description, the words ‘red circular border’ play an important role in identifying the prohibition sign. In comparison, ‘blue’ and ‘yellow’ are remarkable characteristics of mandatory and warning signs. However, the real-world case presents a difficulty. As shown in Figure 16a,c, some non-standard traffic signs are encompassed by additional information. The extra area has a perturbation on the VLM calculation. Furthermore, the small size, adverse weather conditions and image blur are negative factors of the proposed method. The absence of color information has a considerable impact on the detection result.

## 6. Conclusions

In this work, we explore a vision–language model-based traffic sign detection method to analyze high-resolution drone images. Aiming at reducing the reliance on the discrete image labels, text descriptions become important prior knowledge for guiding the image processing program. The multi-modal learning technique is launched to separately handle high-resolution images and the keyword dictionary. In addition, our program takes advantage of large-scale pretrained networks. The fine-tuning step is saved to realize the rapid deployment. There are three main components. First, a keyword dictionary is built to provide text descriptions. On one hand, we refer to the regulations in the Chinese national standard. On the other hand, the BLIPv2 model is activated to perform image translation. Second, we perform the multi-modal learning to achieve the feature extraction. Based on the encoder network within the CLIP framework, UAV images and text descriptions are independently converted into the high-dimensional feature vectors. Third, the similarity between images and keywords becomes an applicable tool for determining the categories of traffic signs. Cosine distance and the Softmax function are carried out to compare the visual features and text embeddings.

A practical application in Guyuan, China, is used to test the proposed method. A number of 111 UAV high-resolution images are captured to monitor the condition of traffic sign instances. Further examinations are performed on two public datasets, CCTSDB2021 and TT100K. Based on the experiment results, there are two primary advantages to our VLM-based method. (1) The dependency on the human annotations is significantly alleviated by the employment of multi-modal learning. Rather than discrete labels, the text description becomes a feasible means of explaining the characteristics of traffic signs. According to the computation results, the keyword dictionary is beneficial for addressing the domain shift among diverse datasets. (2) With the objective of realizing rapid deployment, there is no fine-tuning step in our program. The pretrained VLM is used to analyze the input images as well as text descriptions. The experiment results indicate that our program has a strong generalization ability and cross-domain adaptation potential. 

As a preliminary exploration of a vision–language model in the context of traffic sign detection, our program contains two important limitations. First, the detection quality is heavily dependent on the color information. The low visibility and image blur have a negative impact on the calculation result. Second, our program classifies the traffic sign instances into six categories. A small number of categories does not convey abundant information or provide substantial guidance for the intelligent transportation system. Therefore, a potential extension of this work could help to improve detection accuracy under adverse weather conditions. It is feasible to apply the prompt engineering technique to create high-quality text descriptions. Furthermore, another future research direction is to introduce cutting-edge multi-modal techniques. A combination of self-supervised learning and traffic sign datasets is an applicable method for understanding task-specific knowledge.

## Figures and Tables

**Figure 1 sensors-24-05800-f001:**
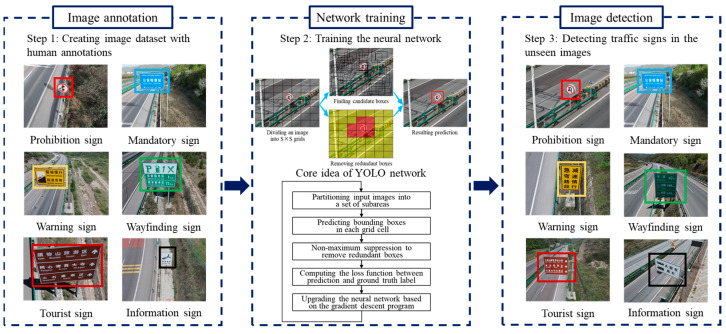
Basic workflow of YOLO-based traffic sign recognition.

**Figure 2 sensors-24-05800-f002:**
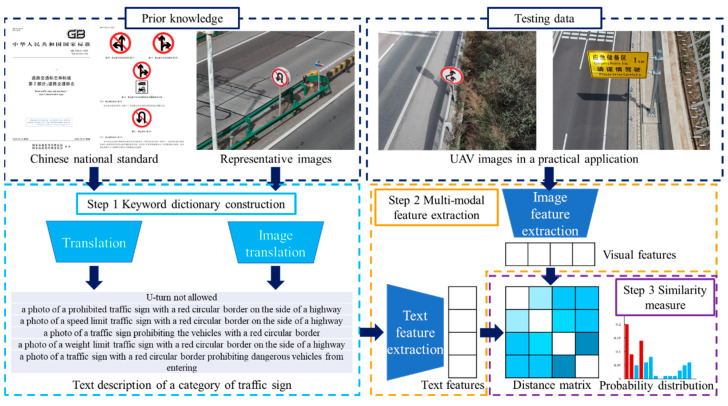
Basic workflow of the vision–language model-based traffic sign detection approach. The Chinese national standard of traffic sign can be found in Reference [37].

**Figure 3 sensors-24-05800-f003:**
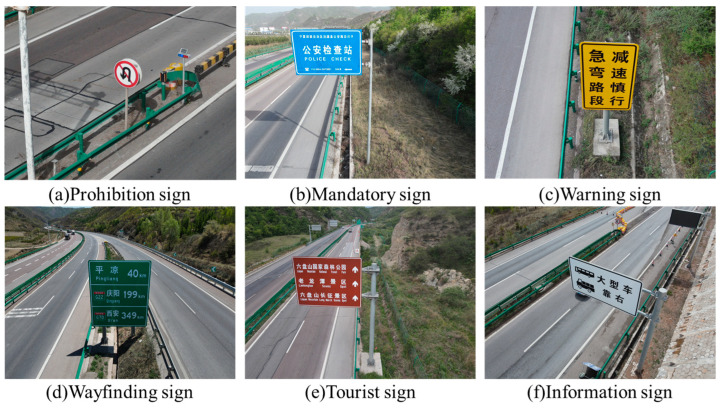
Six categories of Chinese traffic signs.

**Figure 4 sensors-24-05800-f004:**
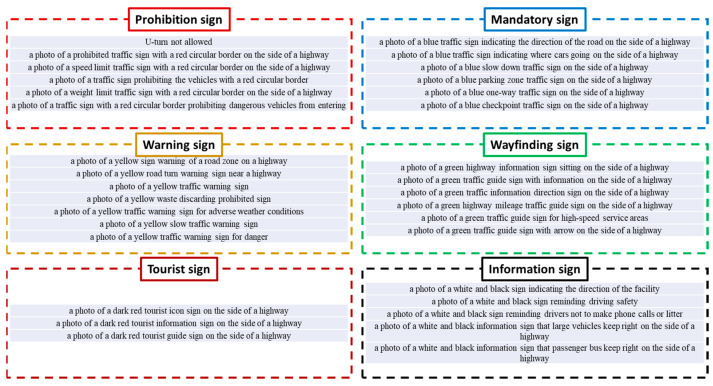
Keyword dictionary constructed by Chinese national standards.

**Figure 5 sensors-24-05800-f005:**
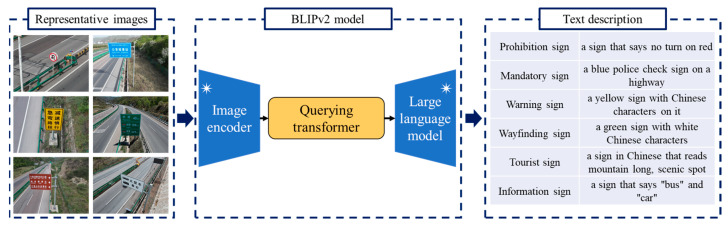
Keyword dictionary constructed by representative images and BLIPv2 model. The mark * indicates that the frozen encoder networks are used by BLIPv2.

**Figure 6 sensors-24-05800-f006:**
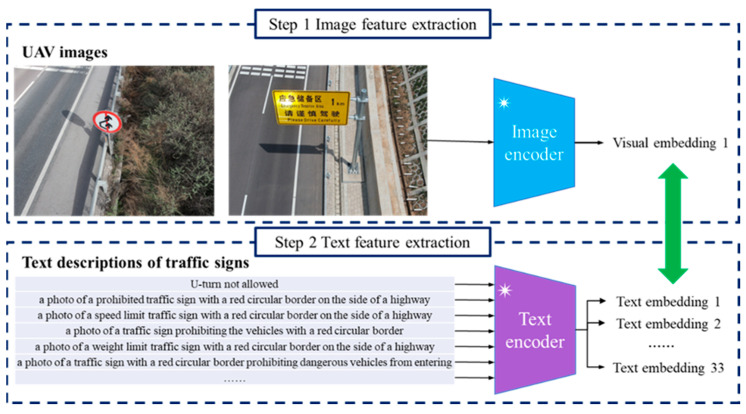
High-dimensional feature extraction with CLIP network. The mark * indicates that we adopt the frozen encoder networks to realize feature extraction.

**Figure 7 sensors-24-05800-f007:**
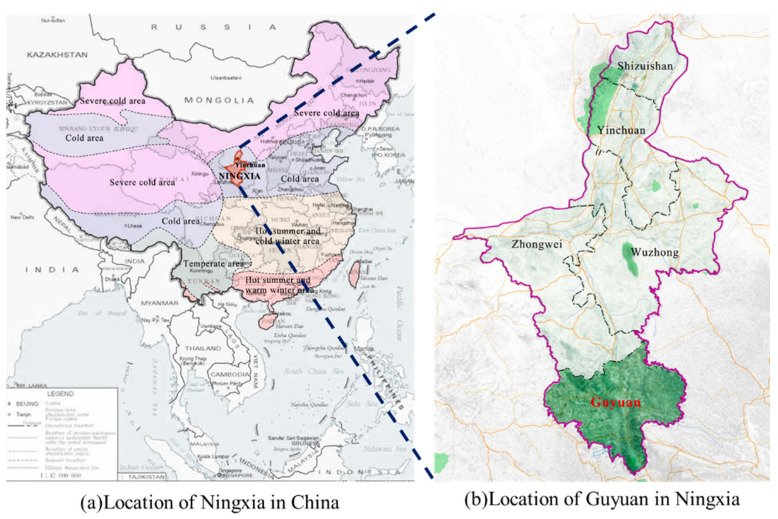
Locations of Guyuan in China.

**Figure 8 sensors-24-05800-f008:**
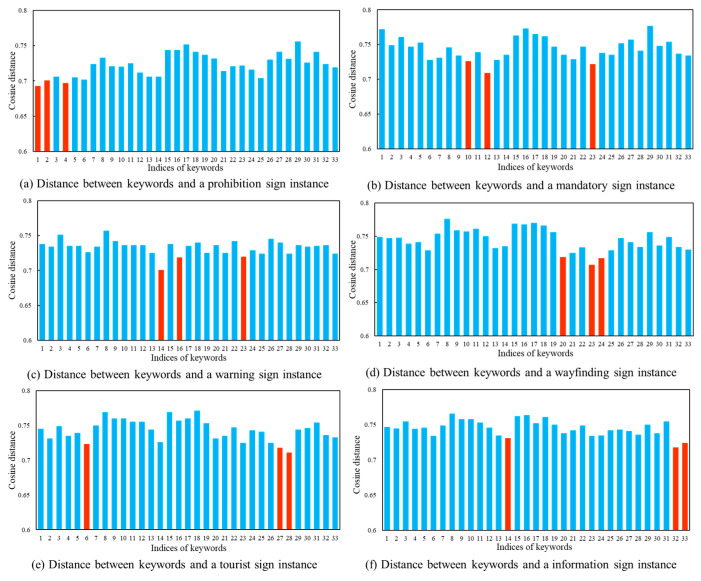
Cosine distance between traffic sign images and keywords.

**Figure 9 sensors-24-05800-f009:**
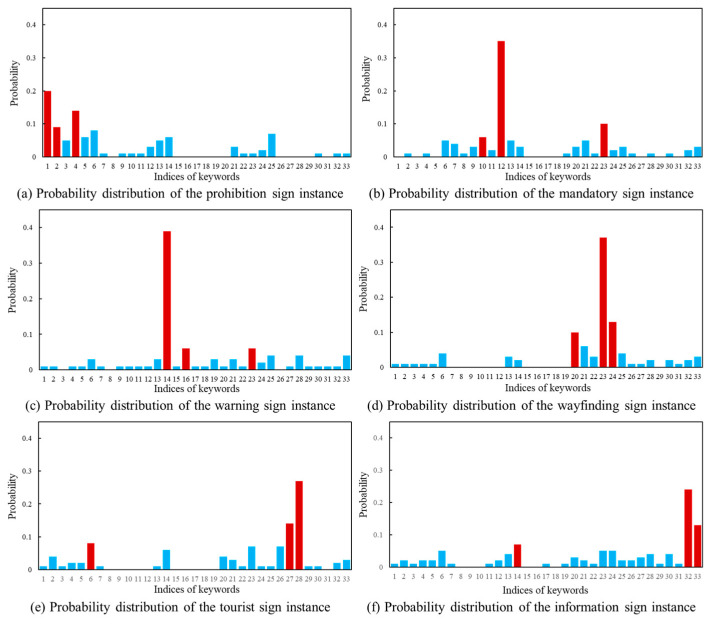
Predicted probability distribution of traffic sign images.

**Figure 10 sensors-24-05800-f010:**
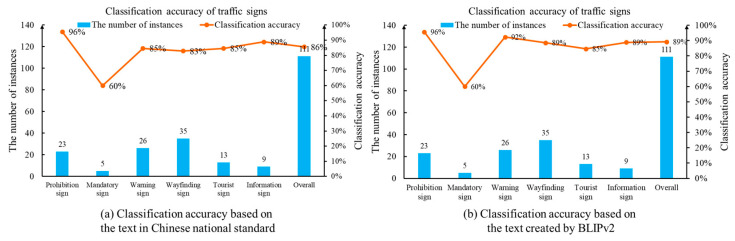
Classification accuracy of traffic sign images based on the proposed method.

**Figure 11 sensors-24-05800-f011:**
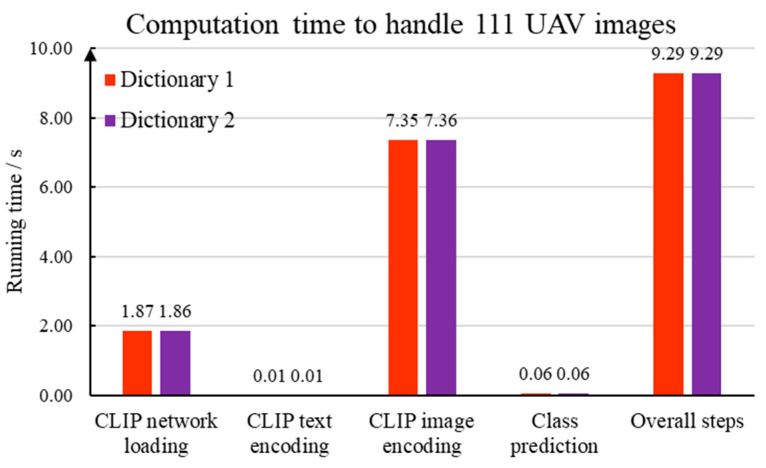
Time consumption of each component in the vision–language model-based method.

**Figure 12 sensors-24-05800-f012:**
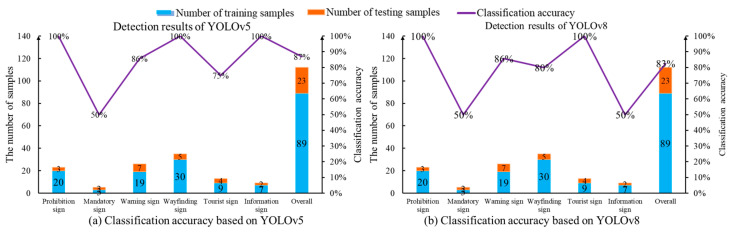
Classification accuracy of traffic sign images based on the YOLO programs.

**Figure 13 sensors-24-05800-f013:**
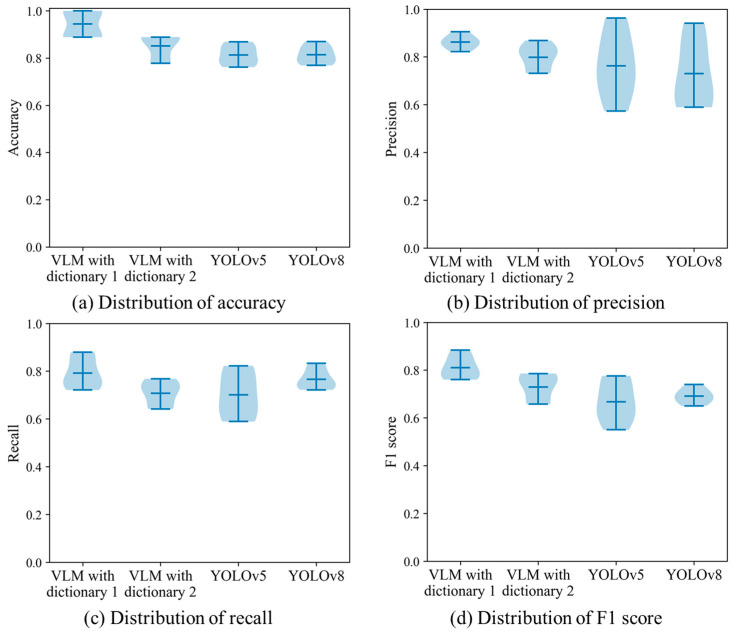
Distributions of four evaluation metrics in the bootstrapping test.

**Figure 14 sensors-24-05800-f014:**
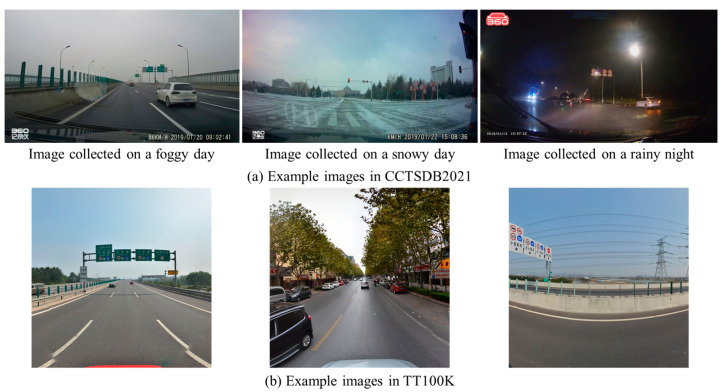
Example images in CCTSDB2021 and TT100K.

**Figure 15 sensors-24-05800-f015:**
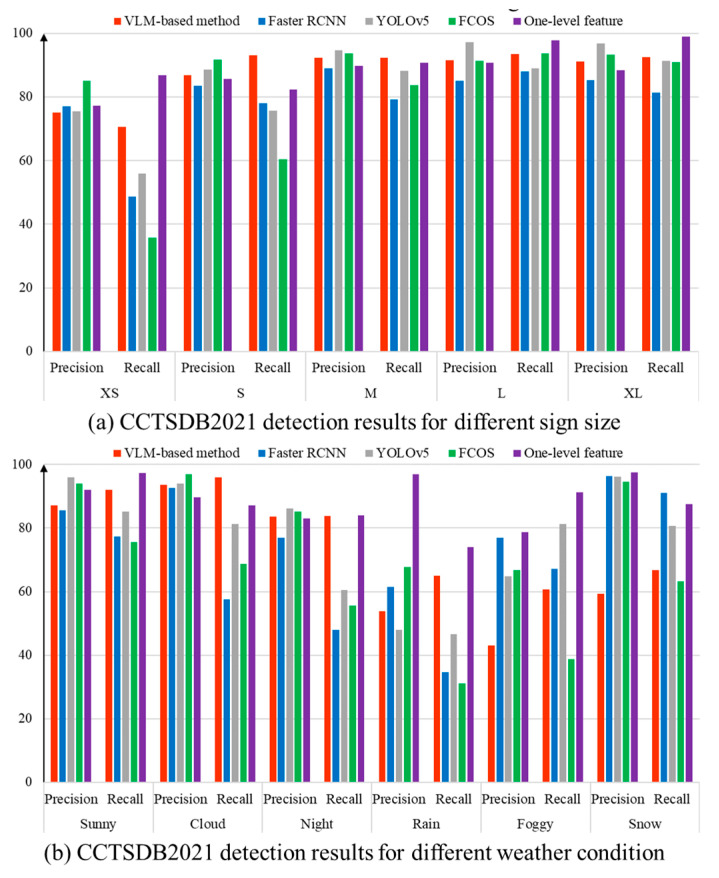
Quantitative evaluation of the CCTSDB2021 dataset.

**Figure 16 sensors-24-05800-f016:**
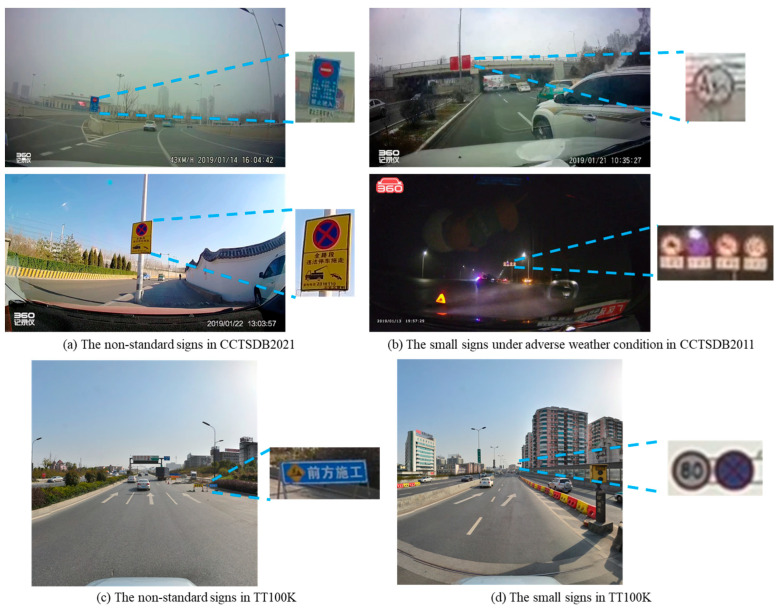
Examples of the false detections in two public datasets.

**Table 1 sensors-24-05800-t001:** Detection results of the UAV images.

Method	Accuracy (%)	Precision (%)	Recall (%)	F1 Score	FPS
VLM with dictionary1	85.6	77.8	85.0	0.80	14.98
VLM with dictionary2	89.2	75.7	78.2	0.75	14.98
YOLOv5	86.9	73.1	80.0	0.76	12.56
YOLOv8	82.6	61.9	76.0	0.68	12.53

**Table 2 sensors-24-05800-t002:** Detection results of our VLM-based method on two public datasets.

Dataset	Accuracy (%)	Precision (%)	Recall (%)	F1 Score	FPS
CCTSDB2021	85.9	82.5	85.4	0.83	45.18
TT100K	92.0	92.2	86.9	0.89	43.67

**Table 3 sensors-24-05800-t003:** Detection results of the CCTSDB2021 with various sign sizes.

Methods	Metrics	XS	S	M	L	XL
VLM-based method	Precision (%)	75.2	86.8	92.4	91.5	91.1
	Recall (%)	70.6	93.2	92.3	93.5	92.6
Faster RCNN [19]	Precision (%)	77.1	83.6	89.0	85.1	85.3
	Recall (%)	48.7	78.1	79.2	88.1	81.3
YOLOv5 [34]	Precision (%)	75.6	88.6	94.7	97.3	96.9
	Recall (%)	55.9	75.7	88.3	89.0	91.3
FCOS [45]	Precision (%)	85.2	91.7	93.8	91.3	93.3
	Recall (%)	35.7	60.5	83.8	93.8	90.9
One-level feature [44]	Precision (%)	77.3	85.7	89.8	90.8	88.5
	Recall (%)	86.8	82.3	90.7	97.9	99.1

**Table 4 sensors-24-05800-t004:** Detection results of the CCTSDB2021 with various weather conditions.

Methods	Metrics	Sunny	Cloud	Night	Rain	Foggy	Snow
VLM-based method	Precision (%)	87.1	93.5	83.7	53.9	43.1	59.3
	Recall (%)	92.1	95.9	83.8	65.1	60.8	66.7
Faster RCNN [19]	Precision (%)	85.5	92.7	76.9	61.4	77.0	96.3
	Recall (%)	77.4	57.6	47.9	34.6	67.1	91.1
YOLOv5 [34]	Precision (%)	95.9	94.0	86.1	47.9	64.8	96.1
	Recall (%)	85.1	81.2	60.6	46.7	81.3	80.7
FCOS [45]	Precision (%)	93.9	97.0	85.1	67.8	66.7	94.5
	Recall (%)	75.5	68.7	55.6	31.1	38.8	63.2
One-level feature [44]	Precision (%)	92.1	89.6	83.1	97.0	78.7	97.5
	Recall (%)	97.4	87.1	84.0	74.0	91.3	87.6

## Data Availability

The data presented in this study are available on request from the corresponding author due to the privacy restriction.

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
