# Peer review of "A Vision–Language Model-Based Traffic Sign Detection Method for High-Resolution Drone Images: A Case Study in Guyuan, China"

_sensors, 2024, doi:10.3390/s24175800_

Round 1

Reviewer 1 Report

Comments and Suggestions for Authors

1. The problem definition and motivation must be clearly stated. Also, please show the main differences between the proposed work and the recently published related works. The contribution of this paper is not organized clearly in the Introduction. This is very important.

2. English writing needs further improvement. The structure of this paper needs to be optimized and the 3. Method section should be highlighted. At present, it is difficult for readers to find the author's scientific discovery or technological innovation.

3. The experiment was conducted only on the UAV dataset constructed by the author. To facilitate comparison among other researchers, it is hoped that this dataset can be made public for free.

4. Please conduct performance evaluation on more public traffic sign detection datasets such as CCTSDB2021, GTSDB, and TT100K. It is worth noting that most of domestic and foreign references indicate that traffic sign detection mainly locates traffic signs and divides them into three categories. However, this paper divides traffic signs into six categories. Please pay attention to this difference when supplementing experiments with other datasets.

5. The title of this paper contains “Traffic Sign Detection Method”, while the title of Section 4 contains “Traffic Sign Classification Task”, which is inconsistent. Is it appropriate?

6. In the evaluation of object detection algorithms, expressions such as Figures 10 and 12 are generally not used. The method proposed by this paper is a detection method, not a classification method. According to Chinese national standard "road traffic signs and markings Part 2: road traffic signs (Standard No. GB5768.2-2009)", warning signs are divided into 44 classes, prohibitory signs are divided into 39 classes, and mandatory signs are divided into 18 classes.

7. Figure 11 shows that the proposed method takes a relatively long time to run. Please provide the metrics such as FPS, FLOPs, and #Param of the proposed method, and compare them with those of other methods.

8. More in-deep analyses, discussion and comparison are needed in Experiments. Competitive results with recent works are needed. It is suggested that the related papers should be cited. For example, DOI: 10.1109/TETCI.2024.3349464; 10.22967/HCIS.2022.12.023; 10.1109/ACCESS.2020.2972338. 

Author Response

First of all, we are grateful to the referee for the review. Each comment is insightful and has a positive effect on improving our manuscript.

Our manuscript is revised in accordance with the comments. We believe these revisions can improve the quality of our manuscript, and hope that it meets the requirements for publication in Sensor.

Please see the detailed responses in the attachment.

Reviewer 2 Report

Comments and Suggestions for Authors

This paper applies Visual Language Models (VLM) to the task of traffic sign detection, demonstrating the potential of multimodal learning in analyzing high-resolution drone images. This approach attempt to not only reduce reliance on manually annotated data but also improves the accuracy and efficiency of detection. Although the paper conducts practical validation in Guyuan City, the scale and diversity of the dataset may not be sufficient to comprehensively validate the generalization capability of the method. The lack of data validation across different environments and conditions might affect the universality of the results.

Comments

1 The data volume in the experimental section is too limited, and the experimental design needs to be restructured. Testing methods suitable for small datasets, such as bootstrapping, could be employed to conduct multiple tests.

2Although the paper mentions comparisons with traditional methods like YOLO, the details of the comparative experiments and the analysis of results may not be sufficiently thorough, failing to fully showcase the strengths and weaknesses of the proposed method.

3It is recommended to collect and validate data in more diverse environments and conditions to enhance the generalization capability and practicality of the method. Specifically, validation in traffic sign detection across different countries and regions could be considered.

4Further comparisons with other advanced methods should be included, along with a detailed analysis of the strengths and weaknesses of each method, to provide more comprehensive experimental results and discussions, thereby enhancing the persuasiveness of the research.

5The text in Figure 3 is unclear.

6The text in Figure 12 is too small to read.

Comments on the Quality of English Language

It is easy to read.

Author Response

Prior to the point-by-point response, we would like to show our gratitude to the reviewer for his/her careful reading and valuable suggestions. Each comment plays an essential role in the manuscript revision.

Our manuscript is revised in accordance with the comments. We believe these revisions can improve the quality of our manuscript, and hope that it meets the requirements for publication in Sensor.

Please see the detailed responses in the attachment.

Round 2

Reviewer 1 Report

Comments and Suggestions for Authors The article has been revised according to the review comments and is even better than expected by the reviewer. The reviewer believes that this article can be published without further modifications.

Author Response

We would like to show our gratitude to the reviewer. Each comment plays an important role in our revision. The quality of manuscript is significantly improved.